# Above-threshold scattering about a Feshbach resonance for ultracold atoms in an optical collider

Milena S.J. Horvath[1], Ryan Thomas[1], Eite Tiesinga[2], Amita B. Deb[1] & Niels Kjærgaard [1]

Ultracold atomic gases have realized numerous paradigms of condensed matter physics, where control over interactions has crucially been afforded by tunable Feshbach resonances. So far, the characterization of these Feshbach resonances has almost exclusively relied on experiments in the threshold regime near zero energy. Here, we use a laser-based collider to probe a narrow magnetic Feshbach resonance of rubidium above threshold. By measuring the overall atomic loss from colliding clouds as a function of magnetic field, we track the energy-dependent resonance position. At higher energy, our collider scheme broadens the loss feature, making the identification of the narrow resonance challenging. However, we observe that the collisions give rise to shifts in the center-of-mass positions of outgoing clouds. The shifts cross zero at the resonance and this allows us to accurately determine its location well above threshold. Our inferred resonance positions are in excellent agreement with theory.

[1] Department of Physics, QSO—Centre for Quantum Science, and Dodd-Walls Centre for Photonic and Quantum Technologies, University of Otago, 730 Cumberland Street, Dunedin 9016, New Zealand. [2] Joint Quantum Institute and Centre for Quantum Information and Computer Science, National Institute of Standards and Technology and University of Maryland, Gaithersburg, MD 20899, USA. Correspondence and requests for materials should be addressed to N.K. (email: niels.kjaergaard@otago.ac.nz)

Resonances lie at the heart of quantum mechanical scattering[1, 2]. They generally result from the occurrence of a metastable state in the encounter between colliding particles and can lead to striking modifications of the scattering cross section. Such metastable states may be established by barriers in the potential landscape of interactions between particles leading to so-called shape resonances. Alternatively, they may take the guise of a quasi-bound molecular state in a given degree of freedom that is coupled to other degrees of freedom into which it can decay. In this scenario, Feshbach resonances arise when the energy of an interacting atom pair in an incident, energetically open channel approaches that of the quasi-bound state. The ability to harness such Feshbach resonance-mediated interactions has been pivotal for ultracold atomic physics for the past two decades. For example, magnetic Feshbach resonances[3] offer a way of tuning both the strength and sign of interactions in ultracold gases and have opened up the opportunity of studying many important fundamental phenomena. These include the creation of bright matter solitons[4], the investigation of the crossover from Bardeen–Cooper–Schrieffer pairing to a Bose–Einstein condensate for strongly interacting Fermi gases[5–12], as well as the coherent formation of ultracold molecules[13, 14].

A Feshbach resonance as a rule modifies the amplitude of elastic scattering where atoms leave in the internal quantum states of the incident channel. Generally, however, the quasi-bound state responsible for the resonance may couple to additional energetically open channels connected to separated atoms pairs in states different to the incoming channel. This describes an inelastic collision. In addition, sufficiently far above the threshold of the incoming channel, the energy of the bound state and thus the resonance position relative to the threshold depend linearly on an applied magnetic field $B$ with a slope defining the differential magnetic moment $\delta\mu$ of the channels. This magnetic field dependence opens up the possibility of tuning the resonance location to the threshold ($E_{res}(B) \to 0$) for atoms in the incident channel and the threshold regime just above it.

Away from threshold, resonant s-wave scattering at relative collision energy $E = \hbar^2 k^2/(2\mu)$ is typically described by the Breit–Wigner formula with an energy-independent width[1]. Here, $\mu = m/2$ is the reduced mass of atoms with mass $m$, and $\hbar k$ is the relative momentum with wavenumber $k$ and the reduced Planck's constant $\hbar$. Near threshold, ($E \to 0$), the coupling between the bound state of the closed channel to the incident collision channel is characterized by a width that depends on the relative collision energy as $\Gamma(E) \propto \sqrt{E}$. In contrast, the partial widths $\Gamma_j^{inel}$ corresponding to the decay from the bound state to inelastic channels $j$ are independent of $E$ (as well as $B$). The s-wave elastic scattering amplitude at magnetic field $B$ is[13, 15, 16]

$$f(E, B) = -\frac{a_{bg}}{1 + ika_{bg}} - e^{2i\delta_{bg}} \frac{a_{bg}\gamma}{E - E_{res} + i\Gamma_{tot}/2}, \qquad (1)$$

where $\Gamma_{tot} = \Gamma(E) + \Gamma^{inel}$, $\delta_{bg} = -\arctan a_{bg}k$ is the background scattering phase shift with scattering length $a_{bg}$, and the reduced width $\gamma = \Gamma(E)/2ka_{bg}$ is independent of both $E$ and $B$. An essential insight is that for $E \to 0$ the regime $\Gamma(E) < \Gamma^{inel} = \sum_j \Gamma_j^{inel}$ will be reached such that $\Gamma_{tot} \approx \Gamma^{inel}$. The resonance position depends on $B$ as $E_{res} = \delta\mu(B - B_0)$, where $B_0$ is the field for which the resonance at threshold occurs. It is worth noting that the total

(elastic plus inelastic) cross section

$$\sigma_{tot}(E, B) = \frac{4\pi}{k} \text{Im}[f(E, B)] \qquad (2a)$$

$$= 4\pi a_{bg}^2 \left[ 1 + \frac{2(E - E_{res})\gamma + \gamma^2}{(E - E_{res})^2 + (\Gamma_{tot}/2)^2} \right] + \frac{2\pi}{k} \frac{a_{bg}\gamma\Gamma^{inel}}{(E - E_{res})^2 + (\Gamma_{tot}/2)^2}, \qquad (2b)$$

is consistent with the optical theorem[15]. The first and second terms of Eq. (2b) correspond to the elastic and inelastic contributions, respectively; in obtaining this expression from Eq. (1) we have assumed $\left| a_{bg}k \right| \ll 1$ so that $e^{2i\delta_{bg}} \approx 1 + 2ia_{bg}k$.

In the realm of ultracold atomic gases Feshbach resonances have, since their inception in these systems[17], predominantly been studied by measuring the loss rate of a trapped, stationary sample as a function of magnetic field. Here, the temperature of the sample would define a characteristic collision energy. Varying the temperature can hence provide some insight into to the energy dependence of the resonance[18–20], but by its nature this method is associated with thermal broadening. Measurements in the energy domain have also been achieved by dissociating Feshbach molecules through a fast and non-adiabatic magnetic field ramp taking the molecular bound state considerably above threshold[21–23]. Here, decay into free atom pairs ensues in what can be described as a half-collision[24]. Obviously, this method relies on the ability to associate atoms into molecules in the first place, which is not generally applicable.

Recently, both experimental and theoretical collider-type approaches that consider energy as a tuning parameter for Feshbach resonances have emerged[25–28]. For example, Gensemer et al.[27] explored resonant scattering behavior with respect to the relative collision energy through use of an atomic fountain that launched ultracold clouds of $^{133}$Cs atoms to collide in free space at a fixed magnetic bias field. In this way, resonances that would overlap at threshold could be resolved. A collider-like configuration was also employed to observe scattering from a Feshbach resonance at a finite energy by Genkina et al. by splitting a trapped cloud of $^{40}$K into two momentum components[28].

In this study, we report on the use of an optical collider[29, 30] based on steerable optical tweezers to explore the $E$ and $B$-dependent scattering of $^{87}$Rb atoms for a Feshbach resonance. In particular, our approach can determine the differential magnetic moment $\delta\mu$ of the resonant quasi-bound level when this strongly couples to outgoing inelastic channels. This setting usually rules out methods relying on associating atoms into molecules[31, 32]. In our experiment, we measure the magnetic field dependence of the total number of particles lost from the two colliding clouds for a range of non-zero energies. We observe a shift of the magnetic Feshbach resonance, but thermal broadening and the $k^{-1}$ fall-off of the inelastic cross section, defined in Eq. (2b), makes the determination of the shifted resonance position exceedingly difficult. However, by instead measuring the center-of-mass positions of the outgoing clouds of unscattered atoms we achieve a dispersively shaped signal, which through its zero-crossing accurately defines the resonant magnetic field at collision energies even far above threshold.

## Results

**System under study.** For our demonstration, we consider a resonance between the $|F = 2, \ m_F = 0\rangle \equiv |2, 0\rangle$, and $|1, 1\rangle$ spin states of $^{87}$Rb located[33, 34] at approximately $B_0 = 0.940(7)$ mT

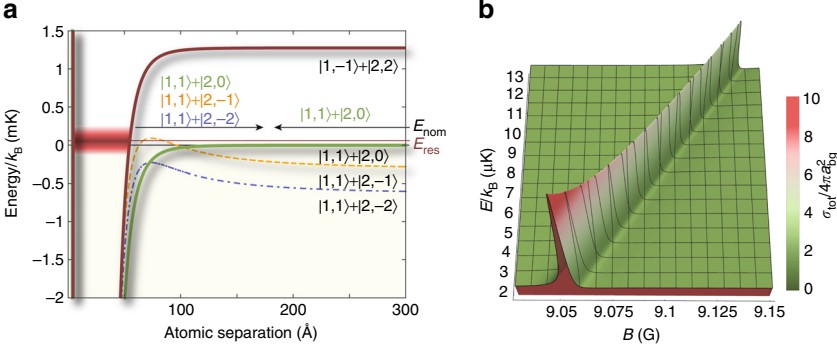

**Fig. 1** Interaction potentials and scattering cross section. **a** Relevant long range *s*-wave and *d*-wave potential curves[52] involved in the Feshbach resonance between incident atoms in the $|1, 1\rangle + |2, 0\rangle$ hyperfine states, labeled according to the free atom states to which they connect. The $|1, -1\rangle + |2, 2\rangle$ potential curve corresponds to a *s*-wave closed channel supporting a quasi-bound state to which the interacting atoms can couple during the collision, provided that the difference in the collision energy of the atoms $E_{nom}$ and the energy of the bound state is small. From the bound state the atoms can elastically scatter back into the *s*-wave entrance channel or decay into either of the two dominant *d*-wave open channels, $|1, 1\rangle + |2, -1\rangle$ and $|1, 1\rangle + |2, -2\rangle$ (inelastic scattering). **b** The total scattering cross section of the 9.040 G resonance as a function of applied bias field and collision energy of the interacting atoms

(9.040 G), which is predicted[33, 35] to have comparable elastic and inelastic widths of $\gamma/\delta\mu = 1.1$ mG and $\Gamma^{inel}/\delta\mu = 2.4$ mG respectively, and a residual magnetic moment of $\delta\mu = 1.998\mu_B$, where $\mu_B$ is the Bohr magneton. Figure 1a shows the relevant interaction potentials for this resonance. The *s*-wave bound state, which is supported by the $|1, -1\rangle + |2, 2\rangle$ potential, is strongly coupled to the inelastic *d*-wave channels $|1, 1\rangle + |2, -1\rangle$ and $|1, 1\rangle + |2, -2\rangle$. The excess energy associated with an inelastic event, as measured in units of the Boltzmann constant $k_B$, is several hundred microkelvin and is gained as kinetic energy as the atoms fly away from the interaction region. Figure 1b presents the expected $E$ and $B$ dependence of the total cross section $\sigma_{tot}$ according to Eq. (2b) for the domain to be covered in our experiments. The Feshbach resonance gives rise to a peak centered on $B_{res} = B_0 + E/\delta\mu$, which shifts linearly with the energy, while its height decreases due to the $k^{-1}$ dependence of the inelastic part of $\sigma_{tot}$.

**Apparatus and experimental procedure.** Our experimental setup has been previously described[36]. Briefly, an ultracold ensemble of $\approx 3 \times 10^6$ $^{87}$Rb atoms in the $|2, 2\rangle$ hyperfine ground-state, at a temperature of $T \approx 1$ μK, is loaded into a far-detuned optical crossed-beam trap. Both the horizontal and vertical beams, making up the dipole trap, are derived from a 1064 nm fiber laser and have waists of $\approx 80$ and $\approx 40$ μm, respectively, at the point of intersection. An acousto-optic deflector (AOD) provides position control for the vertical beam so that the crossing position can be moved along the horizontal guide beam. By toggling the frequency input of the AOD the single cloud of atoms can be split into two, and one of the samples (the projectile) is moved 1 mm along the horizontal guide. Here, we evaporatively cool both samples to below 1 μK by lowering the power of the horizontal trapping beam and initiate a sequence (see "Methods") that prepares the projectile cloud in the $|2, 0\rangle$ state and the stationary cloud (the target) in the $|1, 1\rangle$ state. Using the AOD the projectile cloud is accelerated toward the target cloud to collide at a nominal relative energy, $E_{nom}$, in the presence of a homogeneous magnetic field $B$. At a separation of approximately 80 μm, both vertical beams are switched off, so that the collision occurs while the clouds are confined in the horizontal beam. This procedure gives rise to a 1–3 ms time window prior to the clouds colliding during which atoms evolve ballistically. Approximately 10 ms after the two clouds separate we transfer the atoms in the $|1, 1\rangle$ state to the $|2, 2\rangle$ state using microwave adiabatic passage. Atoms that have not been scattered out of the horizontal guide beam are then detected using standard time-of-flight absorption imaging,

where the sample is exposed to a probe laser beam resonant with the $^{87}$Rb $F = 2 \rightarrow F' = 3$ transition of the D2 line. As remarked previously, atoms involved in inelastic collision events gain several hundred microkelvin in kinetic energy, which far exceeds the depth of the horizontal guide and thus leads to their ejection. In the absence of resonant loss, the imaged projectile clouds contain $\approx 5 \times 10^5$ atoms at a temperature of $\approx 900$ nK, while the target clouds contain $\approx 9 \times 10^5$ at a temperature of $\approx 600$ nK.

**Collisional loss from a Feshbach resonance.** Figure 2 shows loss spectroscopy data for the $|1, 1\rangle + |2, 0\rangle \rightarrow |1, -1\rangle + |2, 2\rangle$ Feshbach resonance as a function of $B$ performed for five collision energies, from $E_{nom}/k_B = 1.7$ μK to 12.0 μK as determined from cloud positions in time-of-flight images. Two distinct loss features emerge from this data. First, as expected from Fig. 1b, we observe a significant peak in atom loss, which shifts with the relative collision energy of the two clouds. This loss adheres to the *dashed gray line* in the figure showing the resonance position predicted by coupled-channels calculations. With increasing energy the feature widens and becomes less distinct. This widening is a result of our acceleration scheme rather than the more fundamental energy dependence of $\Gamma(E)$ at threshold. The spread in energies for the collider depends on the initial cloud temperature and is given by $\delta E = \sqrt{2E_{nom}k_B T}$ (see "Methods"). The *gray shaded area* in Fig. 2 illustrates $\delta E$ associated with a cloud temperature of $T = 600$ nK, characteristic for our experiments. A second non-trivial loss feature occurs around the threshold resonant field $B_0$, and follows the vertical *dash-dotted line* in Fig. 2. We attribute this loss to second-order scattering, where a first elastic scattering event transfers (close to) all of the kinetic energy of a projectile atom to a target atom. Such an event adds a $|1, 1\rangle$ atom traveling along with the $|2, 0\rangle$ projectile cloud. It also adds a $|2, 0\rangle$ atom within the stationary $|1, 1\rangle$ target cloud. Hence, from elastic scattering, secondary $|2, 0\rangle + |1, 1\rangle$ threshold collisions ($E \approx 0$) within both target and projectile clouds ensue, and for the magnetic field $B_0$ a resonant inelastic loss will be encountered. Of course, *s*-wave elastic collisions populate an isotropic halo in momentum space so that secondary scattering occurs for a range of energies and not only at threshold. However, the subset of particles elastically scattered into the projectile and target modes cause the most pronounced loss, since they follow these high-density clouds axially while being radially confined by the horizontal trapping laser beam. Numerical modeling of our collider experiment at $E_{nom}/k_B = 12.0$ μK using the

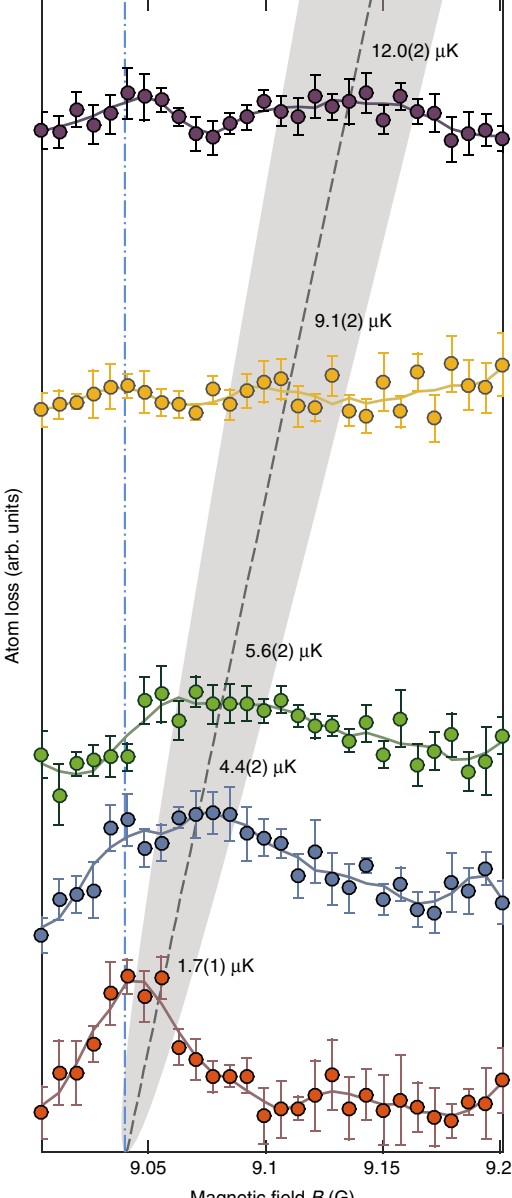

**Fig. 2** Loss spectroscopy in an optical collider. Feshbach loss as a function of magnetic field $B$ for five collision energies $E_{nom}/k_B$ as annotated. Each data set has been vertically offset proportional to $E_{nom}$; a data point (*circle*) represents the mean of typically five loss measurements with *error bars* denoting the standard error of the mean. The *solid lines* are a moving average to guide the eye. The threshold resonance position $B_0$ is indicated by a *dot-dashed blue line*, and the theoretically predicted shift in the resonance position is shown by a *gray dashed line*, surrounded by a *shaded gray region* displaying the expected collision energy spread $\delta E \propto \sqrt{E_{nom}}$ due to the finite temperature of the cloud as discussed in the text and "Methods"

direct simulation Monte Carlo method[30, 37] corroborates our interpretation of the secondary peak.

In conjunction with the inherent energy-dependent resolution of the optical collider, the reduction in the cross section $\sigma_{tot}$ with increasing energy, shown in Fig. 1b, makes a more quantitative comparison between the shifted peak position to theory increasingly challenging. In an attempt to more accurately pinpoint the shifting peak, we next focus our attention to how scattering via a narrow resonance modifies the spatial distributions of the target and projectile clouds.

**Spatial distortions of colliding clouds**. As mentioned above, atoms in the two clouds are allowed to expand freely along the collision axis, $\hat{z}$, for a period of time before they begin to overlap. They remain confined in the transverse radial directions. A one-dimensional, axial model of phase-space distributions in position $z$ and momentum $p_z$ along $\hat{z}$ then suffices to capture the physics at play. Due to the thermal spread $\delta p_z = \sqrt{2mk_BT}$ the initial (Gaussian) phase-space distribution $\rho_p(z, p_z, \tau)$ of the target cloud stretches with time $\tau$ along $\hat{z}$ as shown in Fig. 3a, b during the ballistic expansion[38, 39]. This introduces correlations between the position and momentum of a particle. Those at the front of the target cloud (relative to incident projectile atoms) have mainly negative momentum, while those at the back have positive momentum.

An incident test particle with momentum $p_z = p_p$ interacts resonantly with a particle in the expanding target cloud with momentum $p_z = p_t$, if the difference in their momentum is $p_p - p_t = [4mE_{res}(B)]^{1/2} \equiv \Delta p_{res}(B)$. The resonant interaction removes atoms from a horizontal strip through $\rho_t(z, p_z, \tau)$, with a location that depends on the magnetic field through $E_{res}(B)$—in particular, this strip is centered on $p_z = 0$ when $B = B_0 + p_p^2/(4m\delta\mu) \equiv B_{res}$ as illustrated in Fig. 3d. The loss of particles from the target cloud is imprinted onto its marginal spatial distribution, $n_t(z, \tau) = \int dp_z \rho_t(z, p_z, \tau)$. For $B = B_+ > B_{res}$ (Fig. 3c) interactions involving target atoms with a negative momentum $p_z < 0$, and thus $z < 0$ are predominantly lost due to the correlations introduced by the ballistic expansion. In contrast, $B = B_- < B_{res}$ (Fig. 3e) leads to a predominant loss of target atoms with $p_z > 0$ and $z > 0$. Altogether, this means that the distribution $n_t(z, \tau)$ shifts in the direction of $+\hat{z}$ or $-\hat{z}$, depending on whether $B > B_{res}$ or $B < B_{res}$, respectively.

We can extend these considerations to a thermal ensemble of projectile particles with mean momentum $\overline{p}_p$, and momentum spread $\delta p_z$ (Fig. 4a). Defining now $B_{res}$ in terms of $\overline{p}_p$, Fig. 4b–d illustrate the resulting interactions during transit of the projectile cloud for fields $B = B_+ > B_{res}$, $B = B_{res}$, and $B = B_- < B_{res}$, respectively. As a result of their encounter, the centers of the projectile and target clouds shift oppositely. For $B \gtrless B_{res}$ the target and projectile shift in directions $\pm\hat{z}$ and $\mp\hat{z}$, respectively, and by symmetry the shift is zero for $B = B_{res}$ (Fig. 4c). The shift also becomes zero well away from the resonance.

**Differential magnetic moment from spatially dependent loss**. From the above analysis, we expect to encounter a spatial imprint of the Feshbach resonance on our clouds. We hence turn to an investigation of the center-of-mass positions $\Delta z_{CM}$ of each of the two clouds after the collision, as an alternative to monitoring the total loss in atoms. Figure 5 presents examples of the magnetic field dependence of $\Delta z_{CM}$ as extracted from post-collision absorption images (see "Methods") for both target and projectile for three values of $E_{nom}$. We see that as $B$ is scanned across the resonance, $\Delta z_{CM}(B)$ carries out an oscillation about zero and we take the zero crossing to define the resonance position $B_{res}$. The oscillations observed in the target (Fig. 5a–c) and projectile (Fig. 5d–f) clouds are in antiphase as a result of the complementarity in loss (see Fig. 4b, d). Since the projectile cloud contains fewer atoms than the target cloud its center-of-mass performs a larger excursion. We stress that at the resonant field $B = B_{res}$, no center-of-mass shift is expected in either cloud ($\Delta z_{CM} = 0$) irrespective of the number of atoms in each cloud. This independence makes the determination of $B_{res}$ less sensitive to initial atom number fluctuations as compared to a simple loss measurement. From Fig. 5 a clear upward shift in $B_{res}$ with increasing $E_{nom}$ can be seen in both target and projectile clouds. Figure 6 displays the inferred shifts (see "Methods") from $\Delta z_{CM}(B)$ for our entire data set as well as $B_{res}$ predicted from

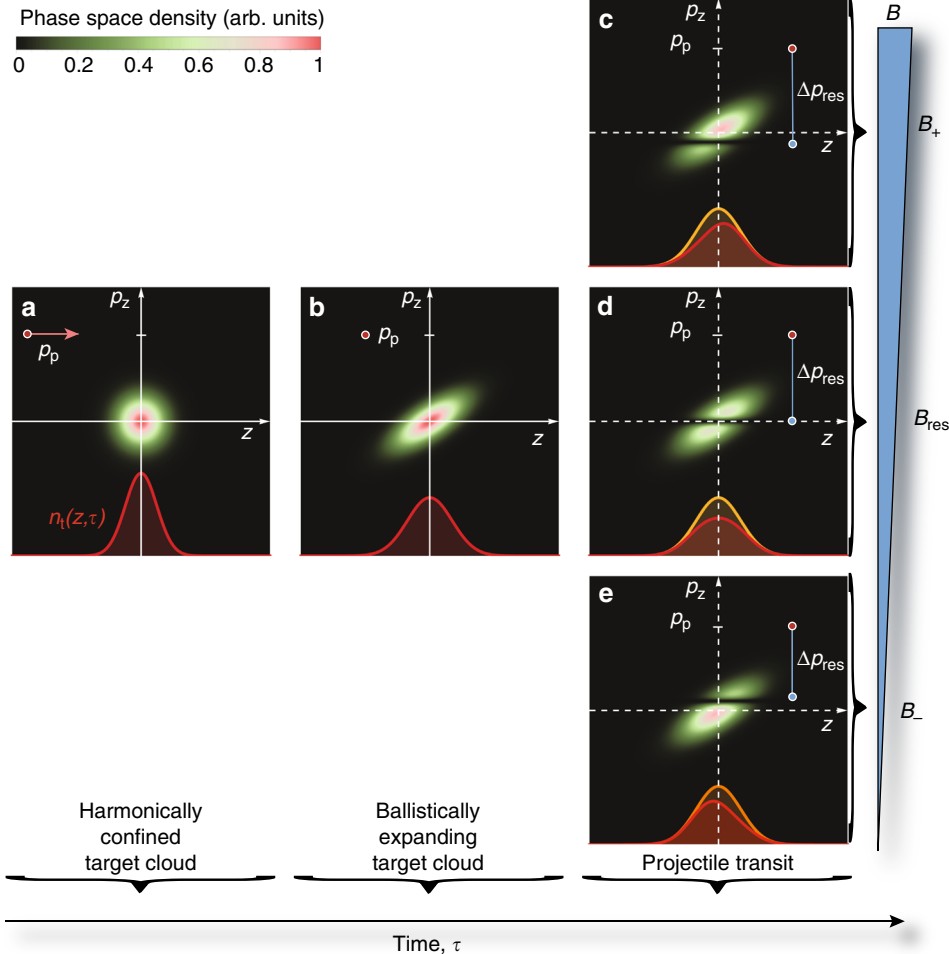

**Fig. 3** Collision between a projectile test particle and a target cloud. Phase-space representations of an incident test particle with momentum $p_p\hat{z}$ moving in the phase-space diagram and colliding with a stationary target cloud. **a** Harmonically trapped target cloud and an incident test particle of momentum $p_p$ as indicated by the *arrow*. **b** Target cloud after some time of ballistic expansion. **c–e** On transit, the test particle can only remove atoms from within a horizontal strip in the phase-space distribution of the target cloud centered at $p_t = p_p - \Delta p_{res}(B)$. Depending on the value of $B$ the strip may be below (**c**), on (**d**), or above the $z$-axis (**e**). In all panels **a–e** the marginal spatial density distributions $n_t(z,\tau)$ of the target cloud are shown at the bottom (*red line*). Panels **c–e** also shows $n_t(z,\tau)$ just before the collision (*orange line*)

coupled channels calculations. The measured $B_{res}$ position agrees well with the predicted linear dependence of the theory, where the rate of the shift is given by $\delta\mu$.

## Discussion

We have investigated the collision of two individually prepared clouds of ultracold atoms about a narrow Feshbach resonance with a significant inelastic component. Previous experiments with colliding clouds successfully analyzed the halos of particles that elastically scattered from incident samples to infer properties about their interaction[23, 24, 28, 40–44]. While a collisional halo also ensues from the process considered in this study[45], its resonant enhancement is difficult to attain due to thermal broadening and the existence of three dominant exit channels. Rather than observing the scattered particles we therefore focused on the atoms remaining in the outgoing clouds following the collision. In this respect, the scheme put forward here is closely related to conventional Feshbach loss spectroscopy, but with the important advantage of adding the collision energy as a tuning knob. While the total number of remaining atoms demonstrated an energy dependence of the resonance position, we found that $B_{res}(E)$ could be inferred with significantly improved accuracy

from the shift in center-of-mass positions of the clouds. This shift is relatively immune to fluctuations in initial atom numbers and is introduced by an imprint of the Feshbach resonance on the spatial atomic distribution. The occurrence of a spatial imprint hinges, crucially, on a position-momentum correlation introduced by ballistic expansion of the clouds prior to collision.

The analysis through shifts in center-of-mass positions opens up a unique way of analyzing the above-threshold behavior for narrow resonance features. In this study, we applied it to a narrow magnetic Feshbach resonance, but the approach is quite generally applicable and could, for example, be used to locate high-$\ell$ shape resonances, which tend to have widths that are very small compared to the resonant collision energy. Our method notably overcomes limitations set by thermal broadening of the collision energy. Such broadening becomes prominent in acceleration schemes[46] where the absolute energy spread of the projectile ensemble increases as it is being accelerated; in particular, our optical collider, where each particle of the projectile cloud accrues the same momentum gain, has these characteristics.

The optical collider scheme presented in this article constitutes an advancement toward directly observing the quintessential $\propto\sqrt{E}$ energy-dependent scaling of the elastic width of an *s*-wave

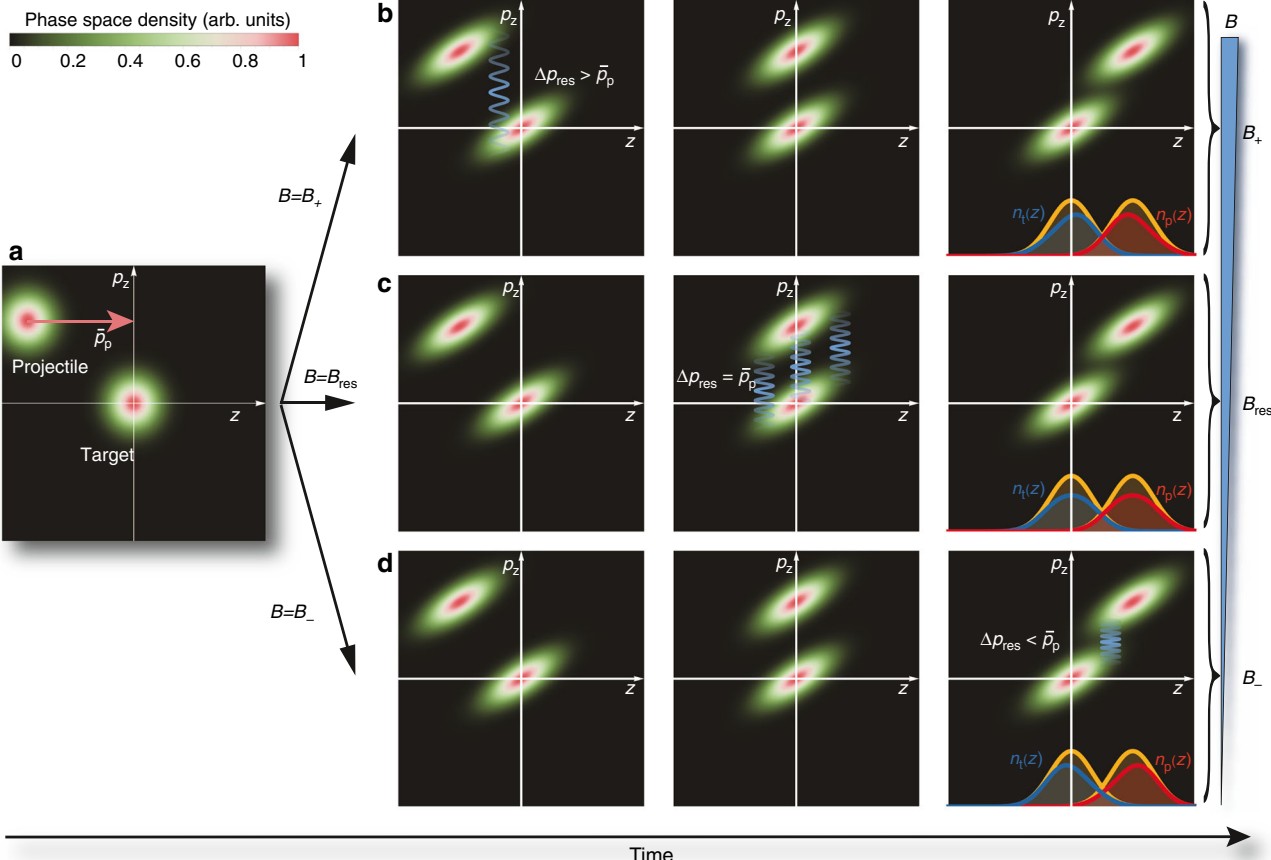

**Fig. 4** Collisions between ballistically expanding projectile and target clouds. Phase-space representations of an ensemble of projectile atoms transiting a stationary target cloud subject to resonant Feshbach loss. Resonant interactions between groups of particles in the projectile and targets clouds are indicated by *wavy blue lines* and happen whenever a projectile particle arrives at the position of a target particle and their momentum difference is $p_p - p_t = \Delta p_{res}$. The actual value for the resonant momentum difference depends on $B$ as $\Delta p_{res}(B) = \sqrt{4mE_{res}(B)}$; in particular, $B_{res}$ defines the field for which $\Delta p_{res} = \bar{p}_p$—the mean momentum of the projectile cloud. **a** Target and projectile atoms are initially confined by two harmonic potentials; the projectile cloud moves with a mean momentum $\bar{p}_p$ toward the target. By switching off the harmonic confinement, the atomic clouds expand ballistically before colliding. **b** For a magnetic field $B = B_+ > B_{res}$, $\Delta p_{res} > \bar{p}_p$ and loss is resonantly enhanced at early times of the transit and occurs at the head-end of the projectile cloud and the $z < 0$ part of the target. **c** For $B = B_{res}$, $\Delta p_{res} = \bar{p}_p$ (by definition) and atoms with momentum difference equal to $\bar{p}_p$ will undergo resonant loss. This condition is fulfilled uniformly throughout the clouds at the point in time where they coincide. **d** For $B = B_- < B_{res}$, $\Delta p_{res} < \bar{p}_p$ and resonant collisions predominantly happen toward the end of the transit and lead to loss at the tail-end of the projectile cloud and the $z > 0$ part of the target. The rightmost frame of **b**–**d** shows how the marginal spatial density distributions $n_t(z)$ and $n_p(z)$ of the target cloud (*blue line*) and the projectile cloud (*red line*), respectively, distort from their shape in absence of resonant loss (*orange line*)

Feshbach resonance at threshold—an effect that in the current experiments still remains too small to be observed. For $^{87}$Rb, a more favorable candidate might be sought in the collisions between pairs of atoms in $|1, 1\rangle$, which is the absolute ground state and therefore not prone to two-body inelastic decay and furthermore has a magnetic Feshbach resonance at 1007 G of a reasonable width ($\gamma/\delta\mu \approx 200$ mG)[47]. More generally, and by extending scattering measurements beyond the threshold regime, the functions parametrizing multichannel quantum defect theory as introduced by Mies[48] could be directly mapped out[49] from careful measurements of resonance widths and positions. Collider studies of Feshbach resonances opens up the possibility of addressing resonances overlapping at threshold[27] and has been suggested as a route to provide limits on the time variation of fundamental constants[27, 50, 51]. Furthermore, it provides an encouraging prospect for an alternative method of observing Efimov physics, where the relative energy dependence of the Efimov state is exploited instead of, as conventional, the dependence on the scattering length through $B$[26]. In conclusion, we believe our optical collider scheme hold considerable promise to be broadly applied.

## Methods

**State preparation and collision procedure.** The two $^{87}$Rb ensembles, separated by 1 mm in the horizontal plane, are exposed to a magnetic field gradient creating a position-dependent Zeeman shift. This enables site selective preparation of the clouds in the $|2, 0\rangle$ and $|1, 1\rangle$ hyperfine states, respectively, via an adiabatic rapid passage procedure using microwave fields. Subsequently, the projectile cloud is moved to a position 0.1 mm from the target cloud, from where it ultimately will be accelerated. For the collision process the gradient field is replaced by a homogeneous bias field. This homogeneous field is first ramped to a value close to the threshold resonant value $B_0$ causing remaining $|1, 1\rangle$ impurities in the $|2, 0\rangle$ cloud and $|2, 0\rangle$ impurities in the $|1, 1\rangle$ cloud to leave the traps through resonant Feshbach loss. After 30 ms the bias field is ramped to its final value $B$ and allowed to stabilize before the beginning of the collision. The magnetic field has a stability of 0.13 mG over 3 days.

**Energy spread in the optical collider.** We consider the collision between a projectile and target cloud at a nominal relative collision energy $E_{nom}$. A particle of momentum $\mathbf{p}_p$ in the projectile cloud collides with a particle with momentum $\mathbf{p}_t$ in the target cloud at relative energy $E = (\mathbf{p}_p - \mathbf{p}_t)^2/(4m)$, where $m$ is the mass of the particles. We assume that the momenta of particles in the target cloud follow a Maxwell–Boltzmann distribution

$$f_t(\mathbf{p}) = \frac{1}{(2\pi m k_B T)^{3/2}} \exp\left[-\frac{\mathbf{p}^2}{2m k_B T}\right], \tag{3}$$

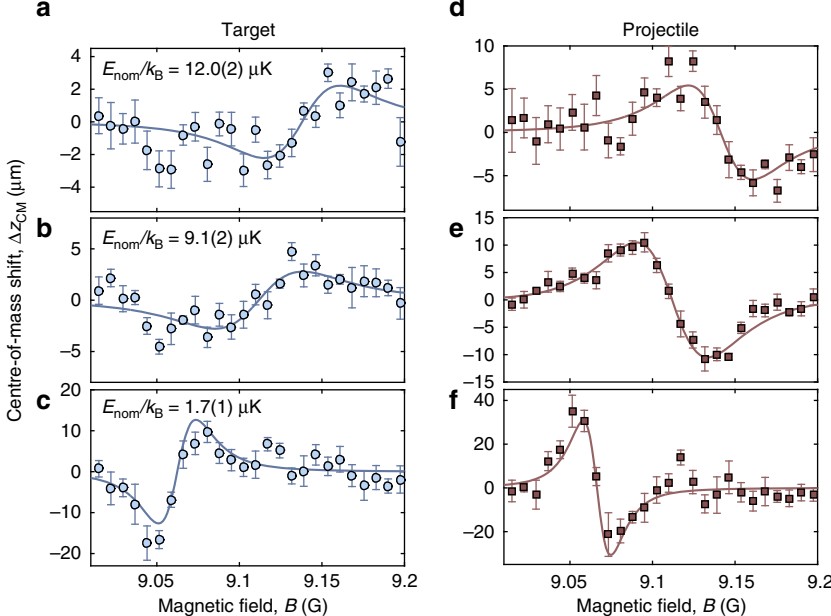

**Fig. 5** Center-of-mass shifts for clouds colliding at a Feshbach resonance. Measured shifts in the center-of-mass $\Delta z_{CM}(B)$ at nominal relative collision energies $E_{nom}/k_B = 12.0(2)\,\mu K$, $9.1(2)\,\mu K$, and $1.7(1)\,\mu K$ for the target cloud (**a–c**, *blue circles*) and the projectile cloud (**d–f**, *red squares*) as a function of $B$. Data points are the mean of five repeats, with the *error bars* corresponding to the standard error. *Solid lines* show pseudo-Voigt fits to the data (see "Methods")

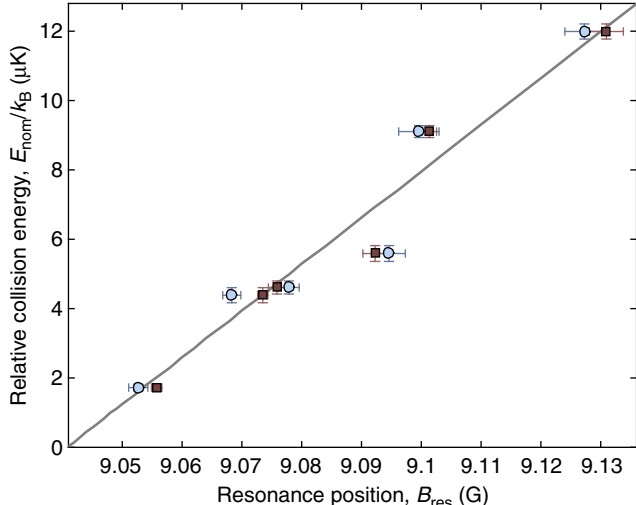

**Fig. 6** Energy dependence of resonant magnetic field. Measured resonance positions (abscissa) vs. collision energy (ordinate) as inferred from the zero-crossing of $\Delta z_{CM}$ for target clouds (*blue circles*) and projectile clouds (*red squares*). The zero-crossings were extracted from weighted curve fits of $\Delta z_{CM}$ using a pseudo-Voigt distribution (see "Methods" and Fig. 5). *Horizontal error bars* denote the $1\sigma$-level confidence bound for fitted values of $B_{res}$. *Vertical error bars* are the standard error for our calibrated collision energies. The *solid line* shows the theoretical prediction for the resonant position

with temperature $T$, whereas the action of the optical collider is to prepare the projectile cloud according to

$$f_P(\mathbf{p}) = \frac{1}{(2\pi m k_B T)^{3/2}} \exp\left[\frac{(\mathbf{p} - \bar{p}_P \hat{\mathbf{z}})^2}{2 m k_B T}\right], \quad (4)$$

i.e., an ensemble traveling along the $z$-axis at mean velocity $\mathbf{v} = \bar{p}_P \hat{\mathbf{z}}/m$, where $\bar{p}_P = \sqrt{4 m E_{nom}}$. Assuming $\bar{p}_P \gg \sqrt{m k_B T}$, we find that the mean collision energy between the clouds is

$$\langle E \rangle = \frac{1}{4m}\left(\langle \mathbf{p}_P^2 \rangle + \langle \mathbf{p}_t^2 \rangle\right) = E_{nom} + \frac{3 k_B T}{2}, \quad (5)$$

with variance

$$\text{var}(E) = \langle E^2 \rangle - \langle E \rangle^2 = \frac{3}{2}(k_B T)^2 + 2 E_{nom} k_B T. \quad (6)$$

Hence, for $E_{nom} \gg k_B T$ the energy spread of the optical collider is

$$\delta E = \sqrt{\text{var}(E)} \approx \sqrt{2 E_{nom} k_B T}. \quad (7)$$

**Modeling center-of-mass shifts of post-collision clouds**. The number of unscattered atoms is extracted from absorption images by fitting the two imaged clouds with Gaussian functions. For both the target and projectile cloud the position of the Gaussian, $\Delta z_{CM}(B)$, is analyzed separately with respect to the applied magnetic field. Figure 5 shows three example data sets of $\Delta z_{CM}(B)$ for a range of bias fields. Each data set (corresponding to a fixed $E_{nom}$) is fitted with the derivative of a pseudo-Voigt profile

$$\Delta z_{CM}(B) = -\eta A \ln(2)\frac{B - B_{res}}{w^2} e^{-\ln(2)(B - B_{res})^2/w^2}$$
$$+ (1-\eta)\frac{2}{\pi}\frac{Aw(B - B_{res})}{\left[(B - B_{res})^2 + (w/2)^2\right]^2} + \Delta z_{bg}, \quad (8)$$

where $0 < \eta < 1$, $\Delta z_{bg}$ is the position of the cloud away from resonance, $w$ is the width, and $A$ is a scaling factor. The data is fitted via a weighted least squares fitting routine, where the weights are determined by the corresponding standard deviations, and $\eta$, $w$, $A$, $B_{res}$, and $\Delta z_{bg}$ are free variables. The field $B_{res}$, corresponding to the inflection point of Eq. (8), varies within the error of the applied bias field when the first (derivative of a Gaussian) or the second (derivative of a Lorentzian) term of Eq. (8) is omitted for the fit.

**Data availability**. The data that support the findings of this study are available from the corresponding author on reasonable request.

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

## Acknowledgements

This work was supported by the Marsden Fund of New Zealand (Contract No. UOO1121). M.S.J.H. conducted her work under a scholarship funded by the New Zealand Tertiary Education Committee through the Dodd-Walls Centre and a University of Otago Postgraduate Publishing Bursary (Master's).

## Author contributions

A.B.D. and N.K. conceived the project. M.S.J.H. and A.B.D. performed experiments with support from R.T. M.S.J.H. analyzed the data with support from R.T. and A.B.D. E.T. provided theory. M.S.J.H. and N.K. prepared the manuscript with input and comments from all authors. N.K. supervised the project.
