## [Peer Review File · Nature Communications]

Reviewers' comments:

Reviewer #1 (Remarks to the Author):

This research demonstrates the use of an optical collider based on steerable optical tweezers to explore the E and B dependent scattering of 87Rb atoms for a Feshbach resonance. In comparison to previous work from this experiment that was focused on the scattering halo, here the authors study the atoms which remain in one of the scattered clouds.

The analysis of the experiment involves a detailed treatment of the expanding clouds including thermal spread. This is not trivial, but it has been done carefully, which is demonstrated by the good agreement of their final result with the theoretical model that involves very precise interaction potentials. This final result is the magnetic moment difference between the collision threshold and the quasi-bound state, which due to the presence of inelastic losses is usually very hard to obtain from other type of cold atom experiments. The paper is written very clearly, it is highly original work and is suitable for publication. However, I have a few remarks that the authors should take into account before this manuscript can be published.

The authors state that investigation of energy-dependent scattering has been very limited with ultracold atom experiments, which often involves only the scattering length. A few recent papers that treat now energy as a tuning parameter have been cited. However, they missed another experiment [Volz et al, Phys. Rev. A 72, 010704 (2005)] which predates the mentioned references.

In this reference, the energy of the colliding atoms probing a shape resonance is tuned via the energy of Feshbach molecules. The magnetic field dependence then controls the energy and lifetime of this d-wave shape resonance. Analytic methods were put in a followup paper [Dürr et al, Phys. Rev. A 72, 052707 (2005)] that comprise expressions for the energy-dependent cross section near a Feshbach resonance including background scattering and inelastic widths, which are very similar to Eq.(2). Moreover, as the detection of shape resonances is also mentioned by the authors of the current manuscript as a particular advantage of their experimental approach, these two papers should be cited as well.

The appearance of a low-energy resonance in Fig.2 is attributed to second-order scattering. However, the explanation is not so convincing. Why would close to all of the kinetic energy of the projectile atom be transferred to a target atom? In a pure head-on collision this would be the case, however s-wave scattering is isotropic, so I would expect all collision energies between 0 and E_{nom} to be present for a secondary collision.

Reviewer #2 (Remarks to the Author):

The paper by Horvath et al. presents theoretical predictions and experimental measurement of the behavior of an atomic Feshbach resonance as a function for relative momentum. Previous experimental results have to date been in the limit approaching zero relative momentum. The experimental results are primarily facilitated by the development of a new technique for determining the location of the resonance. The authors' technique, based on the out-going momentum distribution of the un-scattered atoms, allows them to determine the location of the resonance much more accurately than with traditional techniques such as Feshbach loss spectroscopy and halos, despite thermal and kinetic broadening in the collider used to perform the measurement.

Their measurements showing that the location of the Feshbach resonance varies versus relative momentum match closely with theoretical calculations and have implications for the bandwidth limitations of using a Feshbach resonance to control atom-atom interactions for the purpose of quantum simulation. In addition their novel spectroscopy technique using the center-of-mass positions of the out-going un-scattered clouds could prove a valuable addition to the experimental techniques used to study Feshbach resonances, or other narrow scattering features, such as Effimov states. The authors have demonstrated here that there are cases in which their technique can be significantly more effective than the traditional methods. The stark contrast in feature visibility between figures 2 and 5 makes this very clear. If anything, the authors could stress this point even more throughout the paper.

The Feshbach loss spectroscopy in figure 2 is important to the structure of the result in that it very clearly highlights the resolving power of the center-of-mass based technique used in the rest of the paper, but I find this figure slightly problematic because the two sets of resonance features described in the text are barely visible, if that. The Feshbach loss spectroscopy data is consistent with the described trends, which are confirmed by the later center-of-mass spectroscopy data, but I do not see that it, by itself, constitutes strong proof of Feshbach resonance moving linearly with incident energy. I am unsure whether two spreading loss peaks would be visible at all without the extremely suggestive guides-to-the-eye and theoretical dashed/dot-dashed lines. I believe that this data would be much better presented as motivating the center-of-mass technique by highlighting the drawbacks of the traditional techniques when applied to this particular resonance and high incident energies.

I find the manuscript to be well-organized and clearly written, with just a few small typos.

-)The aside in the abstract noting the conversion from Gauss to Tesla seems entirely unnecessary. Also, grammatically, it is located too far from the magnetic field value which it describes.

-)In the Results section, the sentence: "The Feshbach resonance gives rise to a peak centred which shifts linearly with the energy." Should centred be removed?

-)In the section Apparatus and experimental procedure, the sentence beginning "Both the horizontal and vertical beams... at the point of intersect." Intersection?

-)In the section Apparatus and experimental procedure, the sentence beginning "By toggling the frequency input of the AOD..." is a comma splice.

-)In the Discussion section, the second to last paragraph ends with the phrase "has this characteristics". This is singular, and characteristics is plural.

Reviewer #3 (Remarks to the Author):

This report is for manuscript #17-06530., "Above threshold scattering of atoms about a Feshbach resonance for ultracold atoms in an optical collider," by M. Horvath et al.

The authors describe and experimentally demonstrate a clever method for examining near resonant scattering in ultracold gases, which is of general interest.

In the experiments, two clouds of ultracold atoms in one spin state are initially prepared and separated in a crossed beam optical trap comprising one horizontal beam and one vertical beam. The vertical beam is spatially modulated (use an acousto-optic modulator) to create separated optical

potentials each containing one cloud. Using two microwave frequencies, the clouds are then transferred to two different spin states. Turning off the vertical beams allows the clouds to collide in the remaining horizontal beam (along the z-axis), which also provides radial confinement. By varying the initial displacement of the atoms, the average relative collision energy between is varied. When relative collision energy $\mu/2 v_{rel}^2$ matches the energy mismatch between the incoming channel and the resonant state in the closed channel, atoms are preferentially scattered out of the initial clouds. Here $v_{rel} = (p_{inc} - p_{target})/m$. (the above two statements might be added early for the general audience) After colliding, the incident and target clouds are allowed to expand ballistically, which correlated the z-position and z-momentum, creating a rotated ellipse in phase space. The phase space distribution is altered by the loss of atoms from the resonant relative momentum slices. Instead of monitoring the loss of atoms arising from inelastic collisions, as has been done by several groups, the CM position of each of the clouds is measured. As the initial relative energy or magnetic bias field B is varied, the resonance slices move from below to above $p_z = 0$, causing a corresponding change in CM position, which has a dispersive dependence on B. The CM positions of the target and incident clouds is anti-correlated, further increasing the sensitivity. In this way, the resonant B-Field can be mapped out as a function of initial relative energy, providing much better resolution than simple loss measurements.

The clever general method employed in this paper makes an important step forward in studies of elastic and inelastic collisions with ultracold atoms. Potentially, the method may enable the first direct measurement of the fundamental dependence of the s-wave resonance width on the square root of the relative energy. The method will be generally applicable to studies of higher partial wave resonances. For these reasons, this paper certainly should be published in Nature Communications.

I have a few optional suggestions and comments, which the authors may consider before this paper is accepted for publication.

1)References 5-7 on the BEC-BCS crossover omits two of the earliest references on Fermi gases near Feshbach resonances by the Duke group:

a)O'Hara et al, "Observation of a strongly interacting degenerate Fermi gas of atoms," Science, vol. 298, 2179 (2002), which to my knowledge is the first study of degenerate Fermi gas near a Feshbach resonance and predates all other references.

b)Kinast et al, "Evidence for superfluidity in a resonantly interacting Fermi gas," Phys. Rev. Lett., vol 92, 150402 (2004).

c)It is a bit of a miss-statement to suggest under Fig. 1 that Feshbach resonances have been studied mostly in the zero energy limit: The widely studied unitary limit, is of course the $k a \rightarrow \infty$ case. Do the authors mean low energy compared to the hyperfine coupling energy?

2)It would be helpful to define $\Gamma_{tot} = \Gamma(E) + \Gamma_{inel}$ before equation (1). This makes the discussion regarding $\Gamma(E) < \Gamma_{inel}$ for small E more clear.

3)Under the RESULTS: the last sentence has a peak "Center" which shifts...

4) Δ_{bg} is only $-k a_{bg}$ for small phase background shifts, which is not always the case. I think eq. 1 is probably valid for $\tan \Delta_{bg} = -k a_{bg}$, which is more general.

5)Generally, I found the print in all of the figures to be much too small, especially Fig. 3 and 4: 4 bcd is so small that I did not immediately realize that the marginal distributions were plotted relative to

the orange initial distributions. Very hard to see!

6)On page 4, under the Fig 3 caption, “stationary $|2,0\rangle$ atoms traveling along with the projectile cloud”— stationary and traveling is not clear.

7)Fig. 4 caption: Should read “Target and projectile atoms are initially confined by TWO harmonic potentials.”

Authors' response to Referee reports on the manuscript "Above threshold scattering about a Feshbach resonance for ultracold atoms in an optical collider"

Considered for Nature Communications

Summary

Referee #1 states that *"The paper is written very clearly, it is highly original work and is suitable for publication"* seconded by Referee #2 who states *"I find the manuscript to be well-organized and clearly written, with just a few small typos"*. Finally, in the words of Referee #3 *"The clever general method employed in this paper makes an important step forward in studies of elastic and inelastic collisions with ultracold atoms. Potentially, the method may enable the first direct measurement of the fundamental dependence of the s-wave resonance width on the square root of the relative energy. The method will be generally applicable to studies of higher partial wave resonances. For these reasons, this paper certainly should be published in Nature Communications."*

Point-by-Point response to Referee's remarks.

We are most grateful for the remarks we have received from all three referees. We detail our responses below.

Referee #1

This research demonstrates the use of an optical collider based on steerable optical tweezers to explore the E and B dependent scattering of 87Rb atoms for a Feshbach resonance. In comparison to previous work from this experiment that was focused on the scattering halo, here the authors study the atoms which remain in one of the scattered clouds.

The analysis of the experiment involves a detailed treatment of the expanding clouds including thermal spread. This is not trivial, but it has been done carefully, which is demonstrated by the good agreement of their final result with the theoretical model that involves very precise interaction potentials. This final result is the magnetic moment difference between the collision threshold and the quasi-bound state, which due to the presence of inelastic losses is usually very hard to obtain from other type of cold atom experiments. The paper is written very clearly, it is highly original work and is suitable for publication. However, I have a few remarks that the authors should take into account before this manuscript can be published.

- Referee #1, Remark (1) : *The authors state that investigation of energy-dependent scattering has been very limited with ultracold atom experiments, which often involves only the scattering length. A few recent papers that treat now energy as a tuning parameter have been cited. However, they missed another experiment [Volz et al, Phys. Rev. A 72, 010704 (2005)] which predates the mentioned references. In this reference, the energy of the colliding atoms probing a shape resonance is tuned via the energy of Feshbach molecules. The magnetic field dependence then controls the energy and lifetime of this d-wave shape resonance. Analytic methods were put in a followup paper [Dürr et al, Phys. Rev. A 72, 052707 (2005)] that comprise expressions for the energy-dependent cross section near a Feshbach resonance including background scattering and inelastic widths, which are very similar to Eq.(2). Moreover, as the detection of shape resonances is also mentioned by the authors of the current manuscript as a particular advantage of their experimental approach, these two papers should be cited as well.*

Response (1,1): We are familiar with this very interesting work and our initial omission was based on a judgement that the MPQ dissociation experiments (references put forward by Referee #1) resided in a different category than the cited references which all are “conventional” scattering experiments. In fact, the MPQ authors themselves set the two approaches apart (quoting Volz et al, Phys. Rev. A 72, 010704 (2005)):

by the centrifugal barrier. Note that interference between different partial waves of cold atoms has previously been observed in *scattering* experiments [12,13,18]. Previous molecule *dissociation* experiments, however, observed only outgoing *s* waves.

Nevertheless, we also appreciate that an inclusion of the MPQ work (using dissociation) certainly adds to the story of the energy dependence of a FBR and both of the above references have now been cited. We also included an even earlier MPQ paper as well as one from MIT. To achieve some balance, we further expanded the section to mention that the temperature of a single trapped sample also, in principle, provides a (crude) tuning knob and we have included 3 prototypical examples:

In the realm of ultracold atomic gases Feshbach resonances have, since their inception in these systems [InouyeAndrewsStengerEtAl1998], predominantly been studied by measuring the loss rate of a trapped, stationary sample as a function of magnetic field. Here the temperature of the sample would define a characteristic collision energy. Varying the temperature can hence provide some insight into to the energy dependence of the resonance [Regal2003,Beaufils2009,Baumann2014], but by its nature this method is associated with thermal broadening. Measurements in the energy domain have also been achieved by dissociating Feshbach molecules through a fast and non-adiabatic magnetic field ramp taking the molecular bound state considerably above threshold [Mukaiyama2004,Durr2004,Volz2005]. Here decay into free atom pairs ensues in what can be described as a half-collision [Duerr2005]. Obviously, this method relies on the ability to associate atoms into molecules in the first place, which is not generally applicable. Recently, both experimental and theoretical collider-type approaches that consider energy as a tuning parameter for Feshbach resonances have emerged [Mathew2013,Wang2010, Gensemer2012,Genkina2015]. For example, Gensemer et al [Gensemer2012] explored resonant scattering behavior with respect to the relative collision energy through use of an atomic fountain that launched ultracold clouds of...

• Referee #1, Remark (2) : *The appearance of a low-energy resonance in Fig.2 is attributed to second-order scattering. However, the explanation is not so convincing. Why would close to all of the kinetic energy of the projectile atom be transferred to a target atom? In a pure head-on collision this would be the case, however s-wave scattering is isotropic, so I would expect all collision energies between 0 and E_{nom} to be present for a secondary collision.*

Response (1,2): We have rewritten the manuscript to read

We attribute this loss to second-order scattering, where a first elastic scattering event transfers (close to) all of the kinetic energy of the projectile atom to a target atom. Such an event adds a $|1, 1\rangle$ atom travelling along with the $|2, 0\rangle$ projectile cloud. It also adds a $|2, 0\rangle$ atom within the stationary $|1, 1\rangle$ target cloud. Hence, from elastic scattering, secondary $|2, 0\rangle + |1, 1\rangle$ threshold collisions ($E \approx 0$) within both target and projectile clouds ensue, and for the magnetic field B_0 a resonant inelastic loss will be encountered. Of

Figure R1: RHS: direct simulation Monte Carlo modelling of our experiment for a collision energy of $12 \mu\text{K}$. LHS: Experimental data (same as in Fig. 2 of the manuscript)

course, s -wave elastic collisions populate an isotropic halo in momentum space so that secondary scattering occurs for a range of energies and not only at threshold. However the subset of particles elastically scattered into the projectile and target “modes” cause the most pronounced loss since they “follow” these high density clouds axially while being radially confined by the horizontal trapping laser beam. Numerical modeling of our collider experiment at $E_{\text{nom}}/k_B = 12.0 \mu\text{K}$ using the direct simulation Monte Carlo method corroborates our interpretation of the secondary peak.

Figure R1 shows the result the DSMC modelling we performed. We believe that inclusion of the simulation study in the manuscript is beyond the scope of the current work for which this peak is an irrelevant and unwanted by-product. However, we do feel the secondary peak warrants mentioning as it, in the data for the two highest energies in Fig. 2, is no less pronounced than the primary peak of interest.

Referee #2

The paper by Horvath et al. presents theoretical predictions and experimental measurement of the behavior of an atomic Feshbach resonance as a function for relative momentum. Previous experimental results have to date been in the limit approaching zero relative momentum. The experimental results are primarily facilitated by the development of a new technique for determining the location of the resonance. The authors’ technique, based on the out-going momentum distribution of the un-scattered atoms, allows them to determine the location of the resonance much more accurately than with traditional techniques such as Feshbach loss spectroscopy and halos, despite thermal and kinetic broadening in the collider used to perform the measurement.

Their measurements showing that the location of the Feshbach resonance varies versus relative momentum match closely with theoretical calculations and have implications for the bandwidth limitations of using a Feshbach resonance to control atom-atom interactions for the purpose of quantum simulation. In addition their novel spectroscopy technique using the center-of-mass positions of the out-going un-scattered clouds could prove a valuable addition to the experimental techniques used to study Feshbach resonances, or other narrow scattering features, such as Effimov states. The authors have demonstrated here that there are cases in which their technique can be significantly more effective than the traditional methods. The stark contrast in feature visibility between figures 2 and 5 makes this very clear. If anything, the authors could stress this point even more throughout the paper.

The Feshbach loss spectroscopy in figure 2 is important to the structure of the result in that it very clearly highlights the resolving power of the center-of-mass based technique used in the rest of the paper,

but I find this figure slightly problematic because the two sets of resonance features described in the text are barely visible, if that. The Feshbach loss spectroscopy data is consistent with the described trends, which are confirmed by the later center-of-mass spectroscopy data, but I do not see that it, by itself, constitutes strong proof of Feshbach resonance moving linearly with incident energy. I am unsure whether two spreading loss peaks would be visible at all without the extremely suggestive guides-to-the-eye and theoretical dashed/dot-dashed lines. I believe that this data would be much better presented as motivating the center-of-mass technique by highlighting the drawbacks of the traditional techniques when applied to this particular resonance and high incident energies.

I find the manuscript to be well-organized and clearly written, with just a few small typos.

Aside from five specific points addressed below (typos), the Referee states for Fig. 2 in the manuscript: “I am unsure whether two spreading loss peaks would be visible at all without the extremely suggestive guides-to-the-eye”, which feel this warrants a comment. In Fig. R2 we plot the loss curve for the three lowest energies (1.7, 4.4, and 5.6 μK). It is the same data as in Fig. 2 in the manuscript. We would make the claim that peaks are visible on the RHS panel of Fig. R2, where no guides-to-the-eye are present. The lines in the LHS panel result from applying a Savitzky-Golay filter (order = 3; frame length = 7) to our data which is a well-defined mathematical operation. So while we do agree with the referee that these peaks are not impressively well-defined and less so for higher energies (shown in Fig. 2 but not shown in Fig. R2), we believe a shifting trend to the right can be inferred. Whether or not the shift can be said to be linear or not is certainly debatable and the submitted manuscript, indeed, reads that the lack of resolution “makes a more quantitative comparison between the shifted peak position to theory increasingly challenging. In an attempt to more accurately pinpoint the shifting peak, we next focus our attention to how scattering via a narrow resonance modifies the spatial distributions of the target and projectile clouds.”

Figure R2: Loss curve for the three lowest energies (1.7, 4.4, and 5.6 μK) with and without lines to guide the eye.

We next turn specific points raised by Referee #2

- Referee #2, Remark (1) : *The aside in the abstract noting the conversion from Gauss to Tesla seems entirely unnecessary. Also, grammatically, it is located too far from the magnetic field value which it describes.*

Response (2,1): It is now introduced in the abstract as 0.940 mT(9.040 G). However unnecessary this may seem, we face the dilemma that the field of ultracold atomic physics almost invariably uses Gauss (G) as the unit for the magnetic field while one of the authors (E.T.) has a NIST appointment and “*In accordance with various Federal Acts, the Code of Federal Regulations, and Executive Order 12770, it is NIST policy that the SI shall be used in all NIST publications. When the field of application or the special needs of users of NIST publications require the use of other units, the values of quantities shall first be expressed in acceptable units, where it is to be understood that acceptable units include the SI units and those units recognized for use with the SI; the corresponding values expressed in the other units shall then follow in parentheses.*”. Gauss (G) is not an acceptable unit according to NIST’s definition. Hence, this arguably somewhat awkward solution.

- Referee #2, Remark (2) : *In the Results section, the sentence: "The Feshbach resonance gives rise to a peak centred which shifts linearly with the energy."Should centred be removed?*

Response (2,2): This sentence should read:“The Feshbach resonance gives rise to a peak centred on $B_{\text{res}} = B_0 + E/\delta\mu$, which shifts linearly with the energy..”, and we have fixed this typo.

- Referee #2, Remark (3) : *In the section Apparatus and experimental procedure, the sentence beginning "Both the horizontal and vertical beams... at the point of intersect."Intersection?*

Response (2,3): ✓

- Referee #2, Remark (4) : *In the section Apparatus and experimental procedure, the sentence beginning "By toggling the frequency input of the AOD..."is a comma splice.*

Response (2,4): ✓

- Referee #2, Remark (5) : *In the Discussion section, the second to last paragraph ends with the phrase "has this characteristics". This is singular, and characteristics is plural.*

Response (2,5): ✓

Referee 3

This report is for manuscript #17-06530.,“Above threshold scattering of atoms about a Feshbach resonance for ultracold atoms in an optical collider,” by M. Horvath et al.

The authors describe and experimentally demonstrate a clever method for examining near resonant scattering in ultracold gases, which is of general interest.

In the experiments, two clouds of ultracold atoms in one spin state are initially prepared and separated in a crossed beam optical trap comprising one horizontal beam and one vertical beam. The vertical beam is spatially modulated (use an acousto-optic modulator) to create separated optical potentials each containing one cloud. Using two microwave frequencies, the clouds are then transferred to two different spin states. Turning off the vertical beams allows the clouds to collide in the remaining horizontal beam (along the z -axis), which also provides radial confinement. By varying the initial displacement of the atoms, the average relative collision energy between is varied. When relative collision energy $\mu/2v_{rel}^2$ matches the energy mismatch between the incoming channel and the resonant state in the closed channel, atoms are preferentially scattered out of the initial clouds. Here $v_{rel} = (p_{inc} - p_{target})/m$. (the above two statements might be added early for the general audience) After colliding, the incident and target clouds are allowed to expand ballistically, which correlated the z -position and z -momentum, creating a rotated ellipse in phase space. The phase space distribution is altered by the loss of atoms from the resonant relative momentum slices. Instead of monitoring the loss of atoms arising from inelastic collisions, as has been done by several groups, the CM position of each of the clouds is measured. As the initial relative energy or magnetic bias field B is varied, the resonance slices move from below to above $p_z = 0$, causing a corresponding change in CM position, which has a dispersive dependence on B . The CM positions of the target and incident clouds is anti-correlated, further increasing the sensitivity. In this way, the resonant B -Field can be mapped out as a function of initial relative energy, providing much better resolution than simple loss measurements.

The clever general method employed in this paper makes an important step forward in studies of elastic and inelastic collisions with ultracold atoms. Potentially, the method may enable the first direct measurement of the fundamental dependence of the s -wave resonance width on the square root of the relative energy. The method will be generally applicable to studies of higher partial wave resonances. For these reasons, this paper certainly should be published in Nature Communications.

I have a few optional suggestions and comments, which the authors may consider before this paper is accepted for publication.

- Referee #3, Remark (1a,b) : References 5-7 on the BEC-BCS crossover omits two of the earliest references on Fermi gases near Feshbach resonances by the Duke group: a) O'Hara et al, "Observation of a strongly interacting degenerate Fermi gas of atoms," Science, vol. 298, 2179 (2002), which to my knowledge is the first study of degenerate Fermi gas near a Feshbach resonance and predates all other references. b) Kinast et al, "Evidence for superfluidity in a resonantly interacting Fermi gas," Phys. Rev. Lett., vol 92, 150402 (2004).

Response (3,1a,b): We have added the two references suggested by the referee. We also added references to works of the groups of Salomon, Hulet, and Ketterle (one additional) conducted in the same period.

-
-
- Referee #3, Remark (1,c) : It is a bit of a miss-statement to suggest under Fig. 1 that Feshbach resonances have been studied mostly in the zero energy limit: The widely studied unitary limit, is of course the $ka \rightarrow \infty$ case. Do the authors mean low energy compared to the hyperfine coupling energy?

Response (3,1,c): We concede the point. This paragraph has now been rewritten (see Response(1,1) above) and does not include this statement.

-
-
- Referee #3, Remark (2) : It would be helpful to define $\Gamma_{tot} = \Gamma(E) + \Gamma_{inel}$ before equation (1). This makes the discussion regarding $\Gamma(E) < \Gamma_{inel}$ for small E more clear.

Response (3,2): Rather than moving $\Gamma_{\text{tot}} = \Gamma(E) + \Gamma_{\text{inel}}$ before equation (1), we have moved the “the discussion regarding $\Gamma(E) < \Gamma_{\text{inel}}$ for small E” down in order to accommodate this.

- Referee #3, Remark (3) : *Under the RESULTS: the last sentence has a peak “Center” which shifts. . .*

Response (3,3): see R(2,2) above

- Referee #3, Remark (4) : *δ_{bg} is only $-ka_{\text{bg}}$ for small phase background shifts, which is not always the case. I think eq. 1 is probably valid for $\tan \delta_{\text{bg}} = -ka_{\text{bg}}$, which is more general.*

Response (3,4): We now define $\delta_{\text{bg}} = -\arctan ka_{\text{bg}}$ after equation (1) extending the validity of the expression of the scattering amplitude. Our experiments are however conducted for k sufficiently small that $\tan \delta_{\text{bg}} \approx \delta_{\text{bg}}$ is valid and the expression after the last equal sign of equation (2) makes use of this. We have added a remark stating “in obtaining this expression from equation (1) we have assumed $|a_{\text{bg}}k| \ll 1$ so that $e^{2i\delta_{\text{bg}}} \approx 1 + 2ia_{\text{bg}}k$ ”

- Referee #3, Remark (5) : *Generally, I found the print in all of the figures to be much too small, especially Fig. 3 and 4: 4 bcd is so small that I did not immediately realize that the marginal distributions were plotted relative to the orange initial distributions. Very hard to see!*

Response (3,5): We have scaled up Figures 3, 4 and 5.

- Referee #3, Remark (6) : *On page 4, under the Fig 3 caption, “stationary $|2, 0\rangle$ atoms traveling along with the projectile cloud”— stationary and traveling is not clear.*

Response (3,6): see Response (2,1) above.

- Referee #3, Remark (7) : *Fig. 4 caption: Should read “Target and projectile atoms are initially confined by TWO harmonic potentials.”*

Response (3,7): ✓

Reviewer #1 (Remarks to the Author):

I am satisfied with the answers to my comments. I recommend publication of the current manuscript in Nature Communications.

Reviewer #2 (Remarks to the Author):

This reviewer provided confidential remarks recommending publication.

Reviewer #3 (Remarks to the Author):

The comments by all three referees are quite consistent. The authors have done a thorough job of responding to the constructive criticism of all of the referees. with the suggested changes by the authors, I believe that the paper should be published in its current form.